

# High-Range Resolution Spectral Analysis of Precipitation Through Range Imaging of the Chung-Li VHF Radar

Shih-Chiao Tsai[1], Jenn-Shyong Chen[2], Yen-Hsyang Chu[3], Ching-Lun Su[3], Jui-Hsiang Chen[3]

[1]Department of Environmental Information and Engineering, National Defense University, Taoyuan, Taiwan
[2]Center for General Education, China Medical University, Taichung, Taiwan
[3]Graduate Institute of Space Science, National Central University, Taoyuan, Taiwan

*Correspondence to*: Yen-Hsyang Chu (yhchu@jupiter.ss.ncu.edu.tw)

**Abstract.** Multi-frequency range imaging (RIM) has been implemented in the Chung-Li very-high-frequency (VHF) radar, located on the campus of National Central University, Taiwan, since 2008. RIM processes the echo signals with a group of
closely spaced transmitting frequencies through appropriate inversion methods to obtain high-resolution distribution of echo power in the range direction. This is beneficial to the investigation of the small scale structure embedded in dynamic atmosphere. Five transmitting frequencies were employed in the radar experiment for observation of the precipitating atmosphere during the period between 21 and 23 Aug, 2013. Using the Capon and Fourier methods, the radar echoes were synthesized to retrieve the temporal signals at a smaller range step than the original range resolution defined by the pulse
width, and such retrieved temporal signals were then processed in the Doppler frequency domain to identify the atmosphere and precipitation echoes. An analysis called conditional averaging was further executed for echo power, Doppler velocity, and spectral width to verify the potential capabilities of the retrieval processing in resolving small-scale precipitation and atmosphere structures. Point-by-point correction of range delay combined with compensation of range weighting function effect has been performed during the retrieval of temporal signals to improve the continuity of power spectra at gate
boundaries, making the small-scale structures in the power spectra more natural and reasonable. We examined stratiform and convective precipitations and demonstrated their different structured characteristics by means of the Capon-processed results.

## 1 Introduction

For decades, radars have been used to investigate various phenomena in the Earth's atmosphere, with a detection height ranging from the atmospheric boundary layer at an altitude of hundreds of meters to the ionosphere at an altitude of hundreds
of kilometers. In 1974, a very-high-frequency (VHF) radar with phased array was first employed to observe clear-air turbulences and atmospheric wind fields in the troposphere, stratosphere, and mesosphere (Woodman and Guillen, 1974); which is later termed as MST (mesosphere-stratosphere-troposphere) radar. This type of radar receives the radar echoes scattered or reflected from the atmospheric refractivity irregularities, and it can also detect the echoes from raindrops, ice crystals, and snowflakes through Rayleigh scattering. In the precipitation observations with VHF radars, the terminal
velocities of the rain drops can be estimated from the Doppler spectra of the precipitation and clear air echoes (Fukao et al.,



1985; Larsen and Röttger, 1987; Chilson et al., 1993). Moreover, VHF radars can also be used to examine the height of the melting layer, the sizes and distributions of precipitation particles, and so on (Chu et al., 1991; Chu and Lin, 1994; Su et al., 2009).

VHF phased-array radars for atmospheric observations typically transmit pulse with a finite coverage in range. The range resolution of a pulsed radar echoes can be improved either by using pulse coding technique or by shortening the pulse width. However, a shorter pulse width contains less power and reduces the detectable height. In view of this, Franke (1990) introduced a frequency-hopped technique to the VHF pulsed radar. In its early stage of development, the frequency-hopped technique used only two frequencies alternately for transmission and was valid for a Gaussian-distributed atmospheric layer structure in a range gate. Chilson et al. (2003) implemented the frequency-hopped technique using more than two

frequencies in an ultra-high-frequency (UHF) radar, which is feasible to investigating Kelvin–Helmholtz instabilities and the dynamics of subsidence inversion. The frequency-hopped technique with several frequencies, commonly referred to as range imaging (RIM; Palmer, et al., 1999) or frequency radar interferometric imaging (FII; Luce et al., 2001), has now been handy for remote sensing of the atmosphere. Further investigations of the RIM technique and its applications have been made by many researchers (Luce et al., 2007, 2010, Chen et al., 2016b, Chilson et al., 2001, 2004, Yu et al., 2010, and the references

therein). Among numerous previous studies of RIM, Chilson et al. (2001) and Luce et al. (2007) compared the generalized refractive index profiles estimated from radiosonde data and the echo power (brightness distribution) retrieved from the RIM, and demonstrated that the vertical structures in the troposphere identified by the RIM technique were natural phenomena and not artificial results. In a later study, Chilson (2004) applied time-signal inversion process to the RIM data to obtain temporal signals in subgates and executed the spectral analysis for each subgate, giving high-resolution spectra of the atmospheric

echoes in the range direction. To improve the continuity of the brightness distribution through the range gate boundaries, Chen and Zecha (2009) developed a calibration process for the RIM, and in a later study Chen et al., (2016a) proposed further a procedure of point-by-point range correction for the RIM analysis of precipitation echoes in the time domain. Such a calibration procedure was applied in this study.

VHF radars are more capable of monitoring the atmosphere and precipitation simultaneously than the conventional

weather radars. To examine the respective characteristics of precipitation and atmosphere, separation of precipitation and atmospheric echoes, typically through spectral analysis, is necessary. Usually, we can identify atmospheric and precipitation echoes in the Doppler spectra, and employ the moment method to estimate the atmospheric and precipitation parameters, e.g., echo power, Doppler velocity and spectral width. In the time domain, however, Palmer et al. (2005) used high-pass and low-pass filters to separate atmospheric and precipitation echoes and explored the effects of turbulence on precipitation. In a

different way, Williams (2012) conducted campaign with VHF and UHF radars, and separated hydrometeor motions, to which the UHF radar is sensitive, from the echo spectra of the VHF radar.

Based on the aforementioned progresses in the radar remote sensing of atmosphere using multi-frequency techniques, we made a study of precipitation, using the time-signal inversion process of RIM data as well as the calibration procedures developed by Chen and Zecha (2009) and Chen et al. (2016a). The observed data were collected on August 21 to 23, 2013,



from the Chung-Li VHF radar located in northwestern Taiwan. The time-signal inversion and the calibration process in the

RIM analysis are reviewed in Sect. 2. The experimental setup of the Chung-Li VHF radar and a two-dimensional optical

disdrometer are described in Sect. 3. In Sect. 4, a comparison of RIM-Fourier and RIM-Capon spectral analyses is made. The

procedure of point-by-point range correction was applied to mitigating discontinuities in the range power spectra of the time-

retrieved precipitation and atmospheric echoes. Effectiveness of the point-by-point range correction was verified in quantity,

and different precipitation patterns were examined. Conclusions are drawn in Sect. 5.

## 2 Inversion method and data processing

### 2.1 Time-signal inversion and spectral analysis

Range imaging uses frequency diversity to retrieve the power density (e.g., brightness distribution) as a function of range

(Palmer et al., 1999). It is similar to the coherent radar imaging (CRI), which estimates the power density as a function of

angular location from the echoes received by several receiving channels (Woodman, 1997; Palmer et al., 1998).

According to Chilson (2004), giving $s_n(t)$ as the column vector of temporal signals for N carrier-frequency pulses where

n=1,…, N, the temporal signals at a specific range location can be retrieved by multiplying the original echo signals of

different frequencies with their respective weightings, as expressed by

$$S_j(t) = \sum_{n=1}^{N} w_{jn} s_n(t) \quad , \tag{1}$$

where j indicates the subgate in a range gate specified in the RIM process, $w$ is the weighting matrix ($w_{jn}$ is the components

of $w$) that modulates the phases and amplitudes of radar signals to cause constructive interference at specific positions. A

component of the weighting matrix based on linear approach is the Fourier method, as expressed by

$$w_{jn} = exp(i2k_n r_j) \quad , \tag{2}$$

where $k_n$ is the n-th carrier wavenumber and $r_j$ is the subgate range. Another form of the weighting matrix can be derived

from Capon's method (Palmer et al., 1999;Yu and Palmer, 2001). The Capon method is a self-adaptive signal processing.

According to Palmer et al. (1999), the components of the weighting matrix was derived as

$$w_{jn} = \frac{V^{-1} e_j}{e_j^+ V^{-1} e_j} \quad , \tag{3}$$

where $e_j = [\ e^{i2k_1 r_j} \quad e^{i2k_2 r_j} \ … \ e^{i2k_n r_j}\ ]^T$, and V is the cross spectrum matrix (also known as the visibility spectrum) of the signals

of frequency pairs. Previous studies suggested that the Capon method is more capable of inhibiting the noise than the Fourier





method. Substituting Eq. (3) into Eq. (1) yields the temporal signals of the subgates.

Discrete target signals at different Doppler frequencies cannot be distinguished in the time-domain imaging processing. Therefore, the temporal signals are converted into the frequency domain, usually using Fourier transform, to obtain the spectra for further analysis. For a vertically pointed radar beam, the Doppler frequency shifts of precipitation echoes are generally larger than those of the atmospheric echoes. The moment method can be used to estimate the echo powers, Doppler

frequency shifts, and Doppler bandwidths of the respective echoes in the spectral domain.

The Capon method is a convenient, efficient, and robust procedure for processing the multi-frequency radar data (Yu and Palmer, 2001). Nevertheless, this method is subjected to a limitation in spectral analysis. When the number of carrier frequencies used for RIM is lower than that of the spectral lines of signals (i.e., Doppler velocities of different targets), this method fails to resolve the Doppler velocities effectively and overlooks objects with weaker echo power (or brightness) (Li

and Stoica, 1996; Garbanzo-Salas and Hocking, 2015). In this study, only the echoes from atmospheric refractivity fluctuations and precipitation particles are concerned, and so five frequencies used for RIM to analyze echoes from these two major targets are adequate.

**2.2 Correcting time delay and range-weighting function effect**

Previous studies have demonstrated that properly correcting the time delay and range-weighting function effect on the radar

echoes can improve the continuity of the RIM brightness distribution at gate boundaries (Chen and Zecha, 2009). The time delay of the radar echoes occurs possibly during the propagation of signals in the media and the signal processing in the radar system, which leads to a range error in computation. On the other hand, the range-weighting function contributes different weightings to the brightness values at various range locations, which should be corrected to restore the microscale structures in the radar volume. To determine the time delay of the radar echoes and the range weighting function, Chen and

Zecha (2009) assumed that the RIM-processed brightness values around the gate boundary of two neighboring range gates can be very close to each other after proper correction. Based on this concept, an estimator for computing the mean squared error (MSE) of a pair of brightness values at two neighboring range gates is used:

$$\Delta B = \sum_{i=1}^{N} \frac{\left( B_{1i} - B_{2i} \right)^2}{B_{1i} B_{2i}} \quad , \tag{4}$$





where $B_{1i}$ and $B_{2i}$ are the two sequences of brightness values estimated around the gate boundary and in the overlapped

regions of two neighboring gates. N is the sequence number of brightness values, which are 31 in this study. For example,

the sequences of $B_1$ ($B_2$) are the values in the lower (upper) range gate with respect to the center of the sampling range gate

and an thus be estimated at the range locations between 60 m and 90 m (-90 m and -60 m) for a 150 m sampling gate with an

imaging step of one meter. Iterated computations with different range/time delays and range-weighting functions (Gaussian

form was assumed) are executed in accordance with Eq. (4) to find the minimum value of $\Delta B$ such that the optimal

correcting parameters of range/time delay and range-weighting function are obtained. It is worthy of mentioning that the

Capon method may give extra range weightings to the brightness values and such effect has been included in the range

weighting function determined from Eq. (4).

## 3 Radar parameter specification

The Chung-Li VHF radar is a monostatic pulsed radar with three square antenna arrays arranged in a triangle. Each array

consists of 64 Yagi antennas that is laid out in an 8 × 8 square, and can launch and receive signals both obliquely and

vertically. After Chen et al. (2009) exhibited the first RIM observation using this radar, a large amount of RIM data have

been collected by the same radar (Chen et al., 2016a). The present study employed the data collected between 21 and 23

August, 2013, when the typhoon Trami passed through the Taiwan area. Observations were conducted by vertically

transmitting and receiving radar signals at carrier frequencies of 51.5, 51.75, 52, 52.25, and 52.5 MHz. The following radar

parameters were set: inter-pulse period of 200 μs, pulse width of 1 μs, number of coherent integrations of 128, lowest

sampling height of 1.5 km, sampling step of 1 μs, and number of range gates of 80. The time interval between two data

points at the same frequency was 0.128 sec that is the multiplication of inter-pulse period, the numbers of coherent

integrations and number of carrier frequencies. The range resolution was 150 m corresponding to the 1-μs pulse width, and

the observational height range was between 1.5 and 13.35 km. A two-dimensional optical disdrometer installed next to the

Chung-Li VHF radar site was employed to measure the raindrop size distribution, rainfall rate, fall velocities of raindrops,

and shapes and sizes of the hydrometeor particles. In this study, only the rainfall rate was used to indicate the occurrence of

precipitation during the VHF radar observation.

## 4 Case analysis and discussion

### 4.1 RIM power spectral analysis

On the basis of the RIM analysis procedure proposed by Palmer et al. (1999), the brightness value as a function of

Doppler frequency, $f$, and range, $r$, estimated through the Fourier and Capon methods are expressed, respectively, by

$$B(r, f) = e^{+} V(f) e \quad ,$$  (5)



$$B(r,f) = \frac{1}{e^+ V(f)^{-1} e} \quad , \tag{6}$$

Figures 1b and 1c show two maps of power spectra obtained in accordance with Eq. (5) and Eq. (6) with the same data set.

The imaging step was 15 m, with corrections of using a constant range delay and Gaussian range-weighting function. Figure 1a displays the echo spectra at the original range resolution (150 m), in which clear-air and precipitation echoes with Doppler velocities respectively at around -0.5 m/s and around -8 m/s below the range of about 5 km were observed. As seen, the intensities of the power spectra obtained by the Fourier and Capon methods were very different. The former was much more intense than the latter by about 25 dB on average. Comparing with the original power spectra, the RIM-retrieved power

spectral quality seemed not to be significantly improved, especially with the presence of salient artificial discontinuity in the height distribution of echo intensities at the edges of range gates. In view of this, the time-signal inversion made with Eq. (1) was utilized in the following analysis. Although the temporal signals can be retrieved at any range location with Eq. (1), the range resolution of the retrieved signals is subject to the limitation of the number of carrier frequencies and the radar wavelength.

Figure 2 shows the power spectra of the case shown in Fig. 1 but with the time-retrieved signals. The signal inversion was executed for each range step of 15 m. A comparison between Figures 1 and 2 shows that, irrespective of high degree of similarity between the Fourier-retrieved power spectra, a substantial improvement with an about 20 dB increase in intensity can be seen in the Capon-retrieved power spectra estimated with the time-inversed signals. Nevertheless, both power spectra display marked discontinuities in the retrieved powers at range gate boundaries that are especially evident in the altitudinal

variations of precipitation echoes, in which the constant range delay was employed in correction. To improve the continuity of the retrieved power at range gate boundaries for the precipitation, a point-by-point calibration of range delay was applied, and its effectiveness was also investigated, as addressed in the following section.

**4.2 Improvement of RIM power spectra for atmospheric and precipitation echoes**

Dynamic behaviors of atmosphere and precipitation structures at small scales could be resolved in the RIM-retrieved power

spectra, with very fine imaging range step (several to tens of meters) that is much smaller than the original range gate width (150 m). However, as shown in Figures 1 and 2, the disadvantage of using RIM to process the precipitation echoes is the presence of artificial discontinuities of the RIM-processed power spectra at range gate boundaries, regardless the range delay and range-weighting function effect have been corrected using the procedure used by Chen and Zecha (2009).

In order to mitigate the artificial discontinuity in power spectra at range gate boundaries, the point-by-point correcting

procedure for the range delay and range-weighting function was employed (Chen et al., 2016a) ; the results are shown in Fig. 3. Figure 3a shows the rain rate observed by the disdrometer installed next to the Chung-Li radar site. As indicated, pronounced rain rate occurred during the period between 18.6 h and 19.9 h. Fig. 3b and Fig. 3c are, respectively, enlarged plots of atmospheric and precipitation power spectra that are taken from Fig. 2, in which the constant range delay and



Gaussian range weighting function are used for image calibration. As seen, the discontinuities in the atmospheric power spectra were too vague to identify throughout the altitude range. By contrast, the discontinuities in the precipitation power spectra were so distinct that they can be easily seen at range gate boundaries. Notice that in the calibration approach proposed by Chen and Zecha (2009), the optimal values of range delay and range-weighting function for correction were obtained by assuming a continuity of atmospheric refractivity fluctuations across the gate boundaries. However, this condition may not be valid for the precipitation particles characterized by discrete and discontinuous distribution of the hydrometeors in nature.

The point-by-point correcting procedure, namely, adjustable range delay proposed by Chen et al. (2016a) may be useful to mitigate the discontinuity of precipitation power spectra at gate boundaries, as shown in Fig. 3d. This correcting procedure has been demonstrated to be effective in the time-domain processing of RIM, but not in the frequency domain. Comparing Fig. 3d with Fig. 3c shows that the feature of discontinuity has been improved greatly at gate boundaries. The improvement can be evaluated via the mean square errors (MSE) of brightness values estimated by Eq. (4) with constant and adjustable range delay corrections, as shown in Fig. 4. The left panel of Fig. 4a displays the RIM power spectra of atmospheric refractivity echoes without any precipitation (referring to Fig. 3a). The right panel of Fig. 4a shows the scatter plot of MSEs, in which the abscissa and ordinate denote the values from constant and adjustable range delay corrections, respectively. As shown, the MSEs of adjustable correction were all below 0.2 with a mean of 0.03253, whereas the MSEs of constant correction were widely scattered with a mean of 0.21463. In view of this, the adjustable correction improved the continuity of power spectra at gate boundaries for the atmospheric refractivity echoes. On the other hand, the left panel of Fig. 4b displays the RIM power spectra of the echoes with atmospheric refractivity and precipitation. As shown in the right panel of Fig. 4b, the MSE values estimated from constant correction distributed in a range of 0.1-0.6 with mean of 0.2393 that was slightly larger than that of the atmospheric refractivity case shown in Fig. 4a. This relatively large MSE values were very likely attributed to the existence of precipitation echoes. With adjustable correction, the MSE values were reduced greatly to less than 0.2 with a mean of 0.025443. Based on these examinations, effectiveness of the adjustable correction was confirmed for both precipitation and atmospheric refractivity conditions.

### 4.3 Estimating the spectral parameters through RIM

To validate the time-signal inversion approach, we have executed examinations similar to those made by Chilson (2004). Figure 5 illustrates an example, in which precipitation and atmospheric echoes coexisted. Shown in the panel (a) is the power spectra with original range resolution of 150 m, and the panel (b) is the RIM-Capon-processed power spectra with a range step of 15 m. The panel (b) exhibits several structured atmospheric layers below 4.5 km and an oscillation of Doppler velocities between 6 and 10 km altitude. Moreover, a melting layer appeared in the range interval of 4.5 and 5 km. Doppler velocities of the atmospheric oscillation and the melting layer were computed and shown in the panels (c) and (d), respectively, where the height profiles of the Doppler velocities calculated from the 150 m range resolution ($V_{STD150}$) and the 15 m imaging step ($V_{RIM}$) were compared. As seen in the panel (c), $V_{RIM}$ was in agreement with $V_{STD150}$, consisting with the



study made by Chilson (2004). On the other hand, the melting layer shown in the panel (d) had a larger difference of Doppler velocities between $V_{RIM}$ and $V_{STD150}$, as compared with the panel (c). Several cases of melting layers showed the same features (not presented here).

To realize how effective the RIM processing is, Chilson (2004) compared the echo powers and Doppler velocities calculated from the original and RIM-processed echoes and made a so-called conditional averaging (explained later) in echo power and Doppler velocity. We noted that Chilson (2004) conducted correction of RIM analysis only for range delay to investigate atmospheric echoes. In this study, both range delay and range-weighting function effect on the echoes were corrected, and both atmospheric and precipitation situations were examined.

Because the original range resolution of 150 m and the RIM-processed range step of 15 m do not match, the RIM subgates that are closest to the centers of the original range gates are selected for conditional averaging. Taking the precipitation echo region as an example, the rain echoes with Doppler velocities between about -4 and -10 m s$^{-1}$ and in a height range below the height of 3.5 km were selected to avoid contaminations due to the echoes from ice crystals or supercooled water in and above the melting layer. Conditional averaging of the RIM-estimated Doppler velocities ($V_{RIM}$) for a specified value interval

of the original Doppler velocities ($V_{STD150}$) was computed as follows. The values of $V_{STD150}$ were first divided into 30 equal sections in the velocity interval between -4 and -10 m s$^{-1}$, then the selected $V_{RIM}$ values that fell within one of the sections were averaged together, resulting in an averaged value of $V_{RIM}$ for each of the velocity sections. These are the conditional average of $V_{RIM}$, denoted as $<V_{RIM} | V_{STD150}>$. As discussed by Chilson (2004) in terms of the power density function of random variables, the conditional averaging of $V_{RIM}$ is defined as

$$\left\langle V_{RIM} \left| V_{STD150} \right. \right\rangle = \rho \frac{\sigma_{V_{RIM}}}{\sigma_{V_{STD150}}} \left( V_{STD150} \right) \quad , \tag{7}$$

where $\rho$ is the correlation coefficient of $V_{RIM}$ and $V_{STD150}$, $\sigma_{VRIM}$ and $\sigma_{VSTD150}$ are, respectively, the variances of $V_{RIM}$ and $V_{STD150}$. The slope of $<V_{RIM} | V_{STD150}>$ versus $V_{STD150}$ can be expressed as, provided they are independent of $\rho$, $\sigma_{VRIM}$ and $\sigma_{VSTD150}$,

$$s = \frac{\partial \left\langle V_{RIM} \left| V_{STD150} \right. \right\rangle}{\partial V_{STD150}} = \rho \frac{\sigma_{V_{RIM}}}{\sigma_{V_{STD150}}} \quad , \tag{8}$$

Accordingly, the physical meaning of the slope of less than 1 suggests that the correlation coefficient of $V_{RIM}$ and $V_{STD150}$ be less than 1 or the variance of $V_{RIM}$ be lower than that of $V_{STD150}$, or both. The same scenario is also valid for the original and RIM-processed echo powers (e.g., the reflectivity discussed in Chilson (2004)).

    We show the brightness and conditional averaging of the atmospheric refractivity and precipitation echoes in Fig. 6. The Chung-Li radar data for the period between 18:48:40 and 19:17:20 UT, 21 August 2013, were examined. The left panels of

Fig. 6 show that $P_{RIM}$ and $P_{STD150}$ are linearly correlated, with a slope approximate to 1. It is evident that the $P_{RIM}$ values tend to be systematically smaller than $P_{STD150}$ by about 3 dB (precipitation echoes) and 4 dB (atmospheric refractivity echoes) in average. This is reasonable because of suppression function of the Capon method employed in the RIM analysis. The slopes



of the linear regression lines best fitted to conditional averaging were less than 1 for both atmospheric refractivity (0.78982) and precipitation (0.69426) echoes, as shown in the right panels of Fig. 6. We also calculated the correlation coefficient $\rho$,

the variances $\sigma_{PRIM}$ and $\sigma_{PSTD150}$ from the radar data, and substituted it into Eq. (8) to obtain the slopes of $<P_{RIM} | P_{STD150}>$ versus $P_{STD150}$ for atmosphere and precipitation echoes (e.g., 0.85679 and 0.66983, respectively), which were very close to the fitting slopes. There was a tendency for the data points at lower values of $P_{STD150}$ to deviate to the higher values of $<P_{RIM} | P_{STD150}>$ for both atmosphere and precipitation echoes; This feature is similar to that reported by Chilson (2004), and can be ascribed to noise contamination and poor performance of the Capon method at low signal-to-noise ratio.

245       After validating the processing of conditional averaging for the echo power, we further examined the Doppler velocities for atmosphere and various precipitation conditions. A case is shown in Fig. 7. The Doppler power spectra of the radar data collected between 10:22:25 and 10:28:58 UT on 21 August 2013 are presented in Fig. 7a. As shown, the atmospheric refractivity and precipitation echoes coexisted in this time interval. The former is characterized by very weak mean vertical air motion with Doppler velocities in a range between -2 and 2 m s$^{-1}$, and the latter displays a distinctive feature of a

brightband at around 4.5 km that separate the solid ice particle echoes and liquid rain drop echoes with different terminal velocities. The type of the precipitation in this environment is categorized as stratiform that is very different from the convective precipitation characterized by strong updraft and absence of brightband structure. Figures 7b and 7c display the scatterplots of $V_{RIM}$ versus $V_{STD15}$ (left panels) and $<V_{RIM} | V_{STD150}>$ versus $V_{STD150}$ (right panels) for the atmospheric and precipitation echoes below 3.5 km, respectively. As seen, $V_{RIM}$ and $V_{STD15}$ were generally in agreement with each other. The

values of $<V_{RIM} | V_{STD150}>$ and $V_{STD150}$ also followed a linear relationship and the fitting slopes, calculated from linear regression analysis, were slightly less than one (e.g., 0.94611 and 0.77251, respectively, for atmosphere and precipitation). We calculated again the correlation coefficient $\rho$, the variances $\sigma_{VRIM}$ and $\sigma_{VSTD150}$, from the radar data and obtained the values of 0.95791 and 0.75599 for the slopes of $<V_{RIM} | V_{STD150}>$ versus $V_{STD150}$, respectively, for atmosphere and precipitation, which were very close to the fitting slopes. Note that the plot of $<V_{RIM} | V_{STD150}>$ versus $V_{STD150}$ for the

atmospheric echoes passed through almost the origin, indicating no offsets were introduced through the RIM processing. Moreover, the slope less than one, according to Eq. (8), implying that either the correlation coefficient of $V_{RIM}$ and $V_{STD150}$ was less than one or the variance of $V_{RIM}$ was lower than that of $V_{STD150}$, or both. The reduction in slope can be understood in terms of sampling volumes that have dissimilar range resolutions in original and RIM-retrieved signals. As discussed by Chilson (2004), highly structured layers and rapidly changed velocity field along the range extent possibly lead to a

decorrelation of the parameters derived from a larger (the original) and a smaller (the RIM-processed) sampling volumes. The power spectra shown in Fig. 7a indeed disclose some highly structured layers. Readers can also refer to Fig. 5 to find the height-dependent structure and vertical velocity in both atmosphere and precipitation echoes. In view of this, it is believed that the RIM processing has provided a higher range resolution to detect distinct range-dependent structures in the atmosphere and Doppler velocity field.




Figure 8 exhibits the case of convective precipitation. As seen, prominent updraft with a maximum vertical velocity of about 5 m/s or more was present in the atmospheric refractivity echoes throughout the height range 2-12 km, which varied much more significantly with height than those of stratiform condition. Because of being retarded by the ascending air motion, the altitudinal variation of falling speed of the precipitation particles followed that of the vertical air velocity. Note that there was absence of brightband structure in the power profile of precipitation echoes around the melting level at height

about 4.5 km. In analogy to the case presented in Fig. 7, $V_{RIM}$ and $V_{STD150}$ were in agreement with each other, as illustrated in the left panels of Fig. 8b and Fig. 8c. For the atmospheric echoes, the fitted and estimated slopes, as given in the right panels of Fig. 8b were very close to those in Fig. 7. However, the slopes given in the right panel of Fig. 8c were particularly increased for the precipitation echoes. In the convective environment, the precipitation particles may sufficiently mix with the turbulent air, producing less structured circumstance and giving a slope closer to one after conditional averaging.

Inspecting the power spectra shown in Fig. 8a, the small-scale range-dependent structures were not visible below 4 km, which was different from the case shown in Fig. 7.

   For a more complete investigation, we carried out the conditional averaging of spectral width, $<SW_{RIM} | SW_{STD150}>$. Figure 9 and Fig. 10 show the respective results of the stratiform and convective precipitations. As before, the values of $SW_{RIM}$ were selected for those subgates corresponding to the center of the 150 m-resolution range gates. Without surprise,

the relationship between $SW_{RIM}$ and $SW_{STD150}$ was generally linear for both types of precipitation. The slops of $<SW_{RIM} | SW_{STD150}>$ versus $SW_{STD150}$ were smaller than one, which were in consistent with the consequences offered by the parameters of Doppler velocity and echo power; that is, the RIM-processed power spectra can disclose smaller-scale structures. A worthy of noting is that the fitting slopes (0.49747 and 0.38532) and the theory slopes (0.65525 and 0.31802) in the right two panels of Fig. 9 were much lower than one; by contrast, the fitting slopes (0.88968 and 0.84358) and theory slopes (0.96807

and 0.88325) given in Fig. 10 were closer to one. This difference in slope highlights different characteristics of the structured atmosphere in stratiform and convective conditions, as compared with the results of echo power and Doppler velocity. The convective atmosphere can mix the precipitation particles and air greatly, which produces similar structures resolved from the original and RIM-processed echoes as well as broader and more vibrated spectral widths in both atmosphere and precipitation echoes, and finally results in a larger slope in conditional averaging.

**5 Conclusions**

This study demonstrated an extended application of multi-frequency technique implemented in the Chung-Li VHF radar. The temporal signals at higher range resolution than the original range gate have been retrieved by the Fourier and Capon methods. The retrieved temporal signals were examined in the Doppler frequency domain (e.g., power spectra) to identify the atmosphere and precipitation echoes in view of their different Doppler velocities. The Capon-processed results have been

utilized for study because of the efficient and robust capabilities of the Capon method for atmospheric radar echoes.
   To improve the range continuity of power spectra in the Doppler frequency domain, the calibrations of range delay and range weighting function effect, which have been proved to be necessary in the time-domain analysis of RIM, have been



made. Moreover, an adaptive correction of range delay based on a point-by-point calibration procedure was demonstrated to be feasible for precipitation echoes. After such handling of radar data, the stratiform and convective precipitation types were

examined for the first time with the RIM technique of the Chung-Li VHF radar. We have validated the RIM-Capon capability not only for the atmosphere but also for the precipitation echoes; this was achieved by employing the conditional averaging for echo power, Doppler velocity, and spectra width. Results of the conditional averaging have verified that the RIM-Capon processed power spectra can indicate finer vertical structures in both atmosphere and precipitation regions. Moreover, the convective condition is expected to mix the atmosphere and precipitation particles sufficiently, resulting in

less structured and more turbulent circumstance that have wider spectral width and larger variation in velocity. The RIM-Capon processed power spectra and conditional averaging results have highlighted this scenario.

It is expected in the future that multi-frequency can combine multi-receiver to resolve the spatial distribution of precipitation in more detail. However, radar hardware and data processing technique are needed to be ready for this object.

**Acknowledgements**

The Chung-Li VHF radar is maintained by the Graduate Institute of Space Science, National Central University, Taiwan. We are grateful to Taiwan Typhoon and Flood Research Institute to provide the precipitation data from disdrometer. This research was supported by the Ministry of Science and Technology in Taiwan.

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

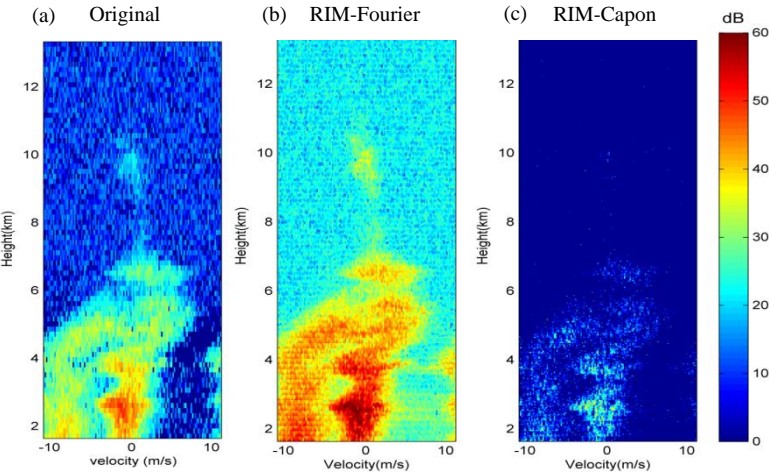

**Figure 1: (a) Power spectra at the original resolution of 150 m. Data time: 19:17:20 UT, 21 August 2013. (b) and (c) are RIM–**
**Fourier and RIM–Capon power spectra, respectively. A constant range delay and range-weighting function corrections have been made in range imaging. Imaging step is 15 m.**

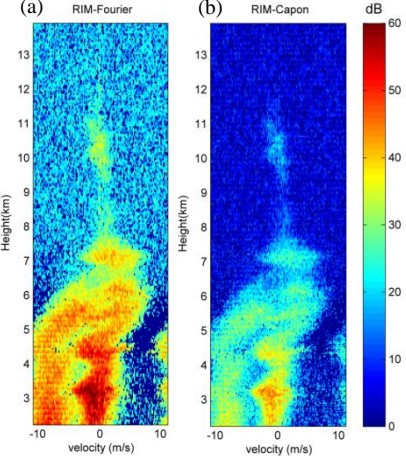

**Figure 2: (a) RIM–Fourier and (b) RIM–Capon power spectra obtained with the temporal signals retrieved from the same radar data in Fig. 1. Imaging step is 15 m.**



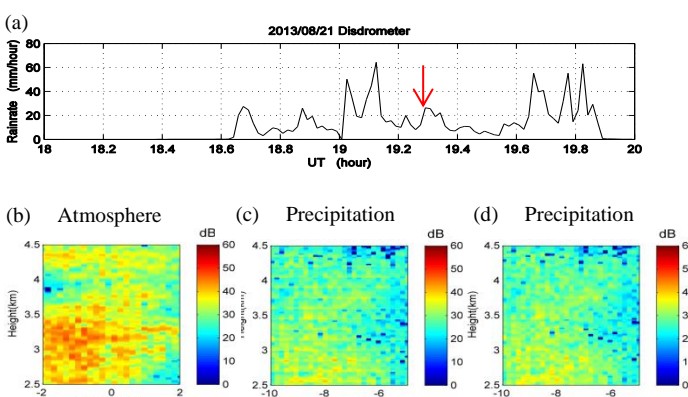


**Figure 3: (a) Rain rate detected by a disdrometer. (b) and (c) are, respectively, power spectra for atmosphere and precipitation echoes at the moment indicated by the arrow in (a). A constant range delay is used for correction. (d) Same as (c) but with point-by-point range delay for correction.**

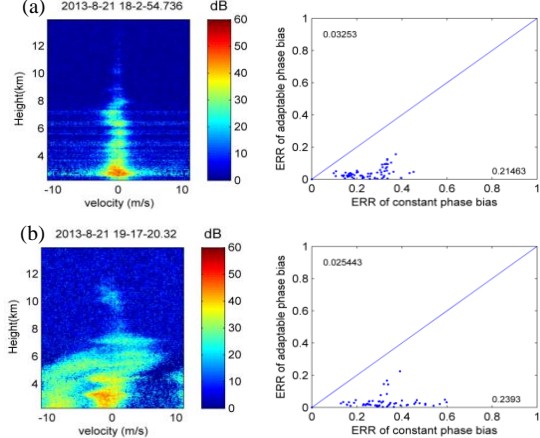

**Figure 4: (a) Rain rate detected by a disdrometer. (b) and (c) are, respectively, power spectra for atmosphere and precipitation echoes at the moment indicated by the arrow in (a). A constant range delay is used for correction. (d) Same as (c) but with point-by-point range delay for correction.**

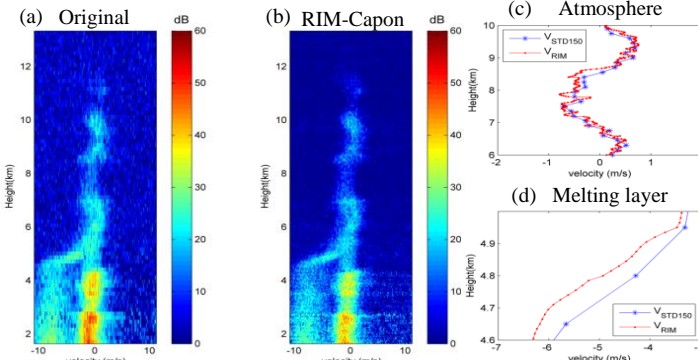





**Figure 5: (a) Conventional power spectra at the original resolution of 150 m. (b) RIM–Capon-processed power spectra at a resolution of 5 m. (c) Height profiles of the Doppler velocities calculated from the 150 m-resolution ($V_{STD150}$) data, and the 5 m-resolution RIM ($V_{RIM}$) data. (d) Same as in (c) but for the melting layer. Data time: 18:51:07 UT, 21 August 2013.**

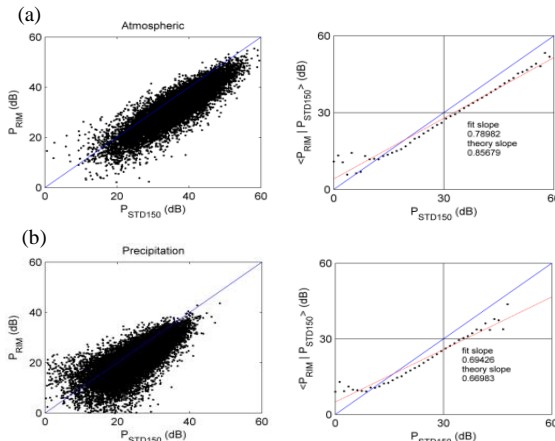

**Figure 6: (a) Results of the conditional averaging $\langle P_{RIM} \mid P_{STD150} \rangle$ for atmospheric echoes in the height range from 1650 to 4000 m AGL. (b) same as (a) but for precipitation echoes. Data time: 18:48:40-19:17:20 UT, 21 August 2013.**

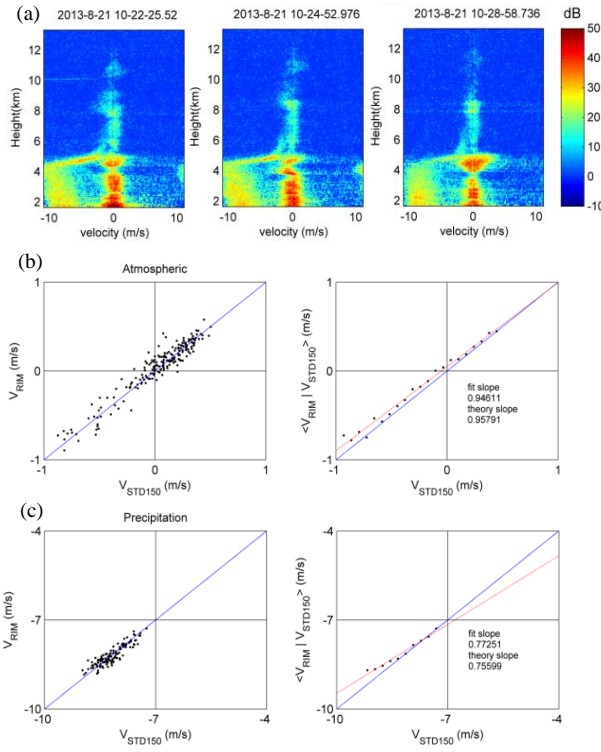

**Figure 7: (a) RIM-Capon processed power spectra during stratiform precipitation. Data time: 10:22:25-10:28:58 UT, 21 August 2013. (b) and (c) are similar to those in Fig. 6a and 6b, but for Doppler velocity.**





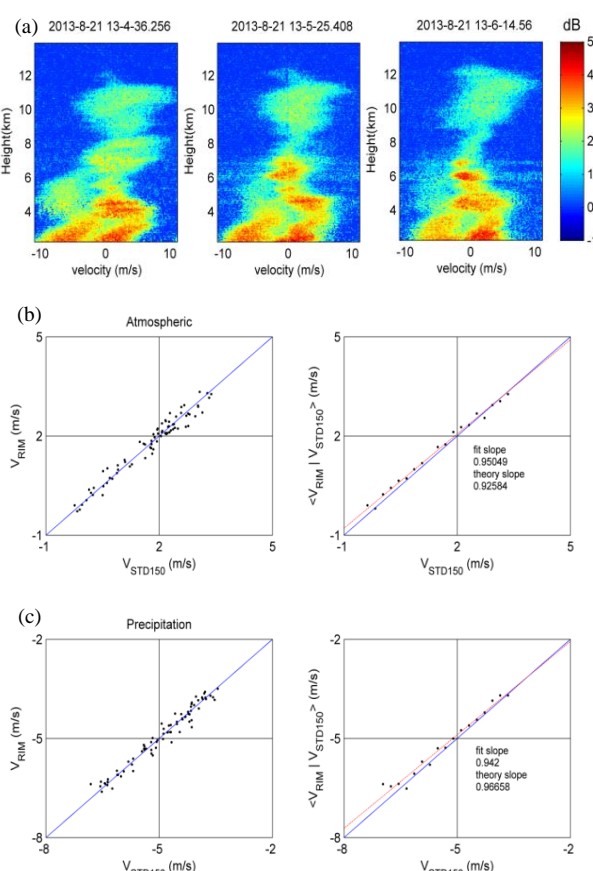

**Figure 8: Same as in Fig. 7, except in convective precipitation condition. Data time: 13:04:36-13:06:14 UT, 21 August 2013.**

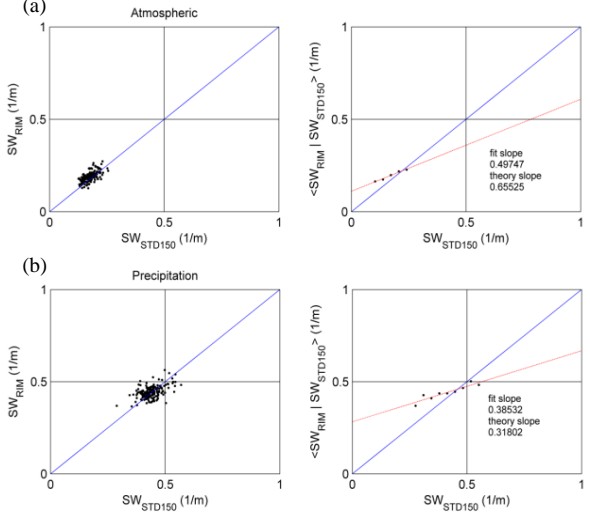


**Figure 9: Results of the conditional averaging like those in Fig. 7b and 7c, except that spectral widths are shown.**



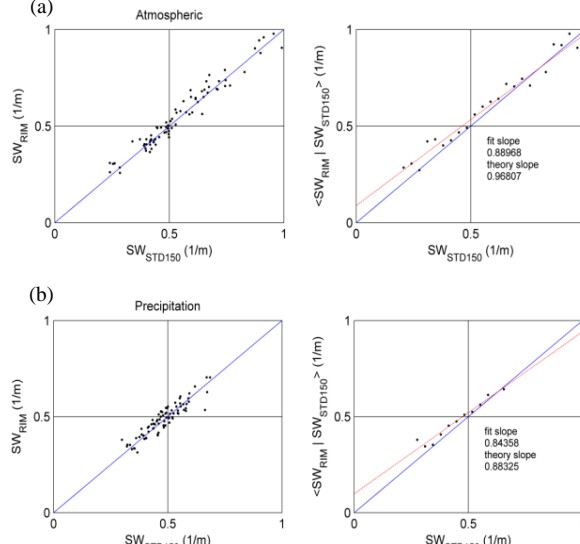



**Figure 10: Results of the conditional averaging like those in Fig. 8b and 8c, except that spectral widths are shown.**