# Peer review of "High-Range Resolution Spectral Analysis of Precipitation Through Range Imaging of the Chung-Li VHF Radar"

_Atmospheric Measurement Techniques, 2017_

## Author Comment (AC1) · 23 Aug 2017

The caption for figure 4 is not correct (it is identical to the caption for Fig. 3 and does not fit with the contents of the figures). The correct is "Figure 4. (a) Left: RIM–Capon power spectra of atmospheric echoes (Data time: 18:02:54 UT, 21 August 2013); Right: scatterplot of the MSEs of echo power with constant and adaptable range delay corrections. (b) same as (a) but for the circumstance with precipitation. (Data time: 19:17:20 UT, 21 August 2013)."

---

## Referee Comment (RC1) · Anonymous Referee #1 · 17 Oct 2017

Review on "High-range resolution spectral analysis of precipitation through range imaging of the Chung-Li VHF radar" by Tsai et al. General comments: Authors discuss the implementation of RIM analysis on precipitation spectra collected with Chung-Li VHF radar. They highlighted the need for point-by-point correction of range delay to ensure the continuity of power spectra at gate boundaries. They also compared two RIM methods, i.e., Capon and Fourier. The paper is, in general, well written and easy to understand. On the flip side, the technique discussed in the paper is not completely new. Even the authors have used it in their earlier studies (for understanding a different region of the atmosphere). What is new is its implementation on precipitation spectra at VHF frequencies. I, therefore, recommend the paper for publication after a moderate

revision. Specific comments: The paper discusses the implementation of RIM on clear-air and precipitation echoes from VHF spectra collected during precipitation. There is no mention about how they segregate these echoes. At times, it is very difficult to segregate them. L218, Why the analysis is restricted to 3.5 km and below. Though authors mention about the effects of radar bright band, it is not clear why to confine only to lower heights. The technique should work at all heights. The authors should highlight clearly what is new in this manuscript. P2, After L61, Include the work of Gan et al. (2015), Radio Sci. Minor comments: L 21, precipitations -> precipitation L 26, turbulences -> turbulence L32, Include Rao et al. (1999), Radio Sci. L40, should be ", which facilitates the investigation of Kelvin-Helmholtz. . . . . ." L64, should be ". . ...collected during 21-23 August 2013 . . . . . ." L115, should be ". . .which is 31 in this study. . ." L117, remove 'an' between 'and' and 'thus' L184, I presume what authors meant was the improvement in continuity not the feature of discontinuity. Please correct the sentence. L253, should be " . . . VSTD150 (left panels). . .." L307, should be ". . .spectral width. . ." L314, should be ". . .objective." Caption of Fig. 4 doesn't match with figures. Needs to be corrected.

---

## Referee Comment (RC2) · Anonymous Referee #2 · 3 Nov 2017

General comments:

An extended application of the multi-frequency range-imaging technique (RIM) implemented in the Chung-Li VHF radar is presented in this manuscript. The RIM technique is used to obtain information about the atmosphere in high range resolution. This method is well known for many years and used be many scientists. Also authors of this manuscript have developed improvement routines and have published various papers about the practical usage of the RIM technique. In this manuscript they show the application of RIM during the simultaneous occurrence of different kinds of precipitation and dynamic background atmosphere conditions, they show the RIM capability

for both echo types, and they present a method and validation for the further improvement of the range continuity. The paper is well written. The authors give a good short introduction, overview and description of the experimental setup. However, because some of the addressed topics in this manuscript (e.g., improved range resolution, comparison Fourier-RIM Capon-RIM, application of RIM to precipitation, range-weighting function effect, point-to-point range correction, conditional averaging) were presented (separately) in earlier papers, the authors should more highlight the new elements and goals of their work. Also additional information about the choice of the short-time cases (within the 3-day-periode), the automatic separation between precipitation and atmosphere echoes, and the restriction to specific height ranges for some studies should be included. A minor revision is recommended.

Specific comments:

L89 previous studies -> include references

L115 "is 31"

L117 "and thus"

L123+ the used time series length (resolution in the frequency domain) for the data analysis is missing.

L144 insert the date and time in the text

L173 "18.6 h and 19.9 h" -> convert it also to HH:MM:SS, because all further times are in this format

L187 "(referring to Fig. 3a)" -> mark it in Fig. 3a

L202 and later: "15 m" -> inconsistent to caption of Fig. 5 ("5 m")

L209 "Several cases of melting layers showed the same features (not presented here)." -> The authors could show Height-Time-Velocity plots for both velocities.

L307 "spectral width"

Figures: Most of the figures are really small and should be enlarged. The advantages of RIM and the improvement methods are not always clear visible in the plots. Sometimes a zoom-in is necessary.

Figure 4a and 4b: Why is it not better to show the errors as function of height? Caption is missing.

Figure 5: "resolution of 5 m" -> inconsistent to text ("15 m")

Figure 6: "height range from 1650 to 4000 m" is not mentioned in the text

Figure 7-10, right panel: Which slope is marked by the red line?

Figures and Text: Five decimal places for the slopes and mean values should not be necessary.